# Small Molecules—Prospective Novel HCMV Inhibitors

**DOI:** 10.3390/v13030474

**Published:** 2021-03-12

**Authors:** Elke Bogner, Anna Egorova, Vadim Makarov

**Affiliations:** 1Institute of Virology, Charité—Universitätsmedizin Berlin, 10117 Berlin, Germany; 2Research Center of Biotechnology RAS, 119071 Moscow, Russia; anna.p.egorova@gmail.com (A.E.); makarov@inbi.ras.ru (V.M.)

**Keywords:** human cytomegalovirus, antiviral therapy, small molecules

## Abstract

Human cytomegalovirus (HCMV), a member of the betaherpesvirinae, can cause life-threatening diseases. HCMV is globally widespread, with a seroprevalence in adults varying from 50 to 100%. HCMV infection is rarely of significant consequence in immunocompetent individuals. However, although immune control is efficient, it cannot achieve the clearance of the virus. HCMV persists lifelong in the infected host and reactivates in certain circumstances. In neonates and in immunocompromised adults, HCMV is a serious pathogen that can cause fatal organ damage. Different antiviral compounds alone or in combination have been used for the treatment of HCMV diseases. In clinical use, mutations in the viral DNA polymerase or the terminase confer resistance to ganciclovir, foscarnet, cidofovir, and letermovir. There is an urgent need to find new well-tolerated compounds supporting different modes of action. The list of novel small molecules that might have anti-HCMV activity has grown in recent years. In this short review, a selection of compounds in clinical trials and novel inhibitors targeting host-cell factors or viral proteins is presented, and their modes of action, described.

## 1. Introduction

Ideally, the treatment of virus-induced diseases would result in total clearance. However, in many cases, including herpesviruses, elimination is impossible because the viruses establish lifelong latency in their hosts, and reactivation during immunosuppression leads to recurrent episodes of disease. Effective vaccines against herpesviruses are not available, and therapeutic drugs often remain the only viable approach. Human cytomegalovirus (HCMV) is a significant pathogen and can cause severe life-threatening infections in immunocompromised, immunosuppressed, and immunonaïve patients. Due to its broad cell tropism, HCMV infection causes a wide range of symptoms and may lead to organ failure [1,2]. Antivirals against HCMV are required, to prevent these severe outcomes.

Recent problems have occurred because most drugs approved for clinical treatment target identical steps in viral replication, leading to a dramatic increase in drug resistance. To date, nearly all the available drugs against HCMV are inhibitors of the viral DNA polymerase. The usage of the same target leads to frequent cross-resistance. Two drugs, ganciclovir (GCV; Figure 1), a nucleoside, and cidofovir (CDV; Figure 1), a nucleotide analogue, need to be activated by phosphorylation. Foscarnet (FOS; Figure 1), a pyrophosphate analogue, does not need activation. GCV and its oral prodrug valganciclovir (Figure 1) are first-line antivirals and used for prevention and for treatment. By contrast, FOS and CDV are mainly used for the therapy of drug-resistant HCMV infections [3]. Clinical use leads to mutations in the viral DNA polymerase and, in the case of GCV, in the phosphotransferase pUL97 that confer resistance [4,5,6]. This happens very shortly after the application of GCV in patients during prolonged therapy [7,8]. Recently, the terminase inhibitor letermovir (Figure 1) was approved for HCMV prophylaxis for hematopoietic stem cell recipients [9]. However, mutations in the HCMV terminase subunit pUL56 [10,11] and, to a lesser extent, in pUL89 [12,13] or pUL51 [14] lead to resistance [6,15,16,17]. Due to the multiple problems caused by the currently available drugs, new antiviral targets with different modes of action are needed. In this review, we describe the development of novel, promising HCMV inhibitors, leading to new perspectives for future therapies. Compounds that are described in other contributions to this Special Issue have not been taken into account.

## 2. Compounds in Clinical Trials

Only a few anti-HCMV compounds are currently in clinical trials. These include two nucleoside inhibitors (filociclovir and brincidofovir) and an inhibitor of the viral protein kinase pUL97 (maribavir).

Zhou et al. [18] synthesized filociclovir (Figure 2), a second-generation analog of 2′-deoxyguanosine containing a methylenecyclopropane moiety. The compound effectively blocks HCMV replication, with EC_50_s of 0.46 µM (Towne) and 0.49 µM (AD169), and is 10-fold more active than GCV [18]. Like other nucleoside inhibitors, filociclovir requires activation by phosphorylation that is mediated by HCMV pUL97 kinase [19]. Furthermore, filociclovir prevents HCMV replication in implant tissues in SCID mice [19] and has a favorable oral bioavailability [19]. As these data demonstrate high efficiency in animal models, the compound was evaluated for the treatment of HCMV infections in humans. In a phase Ib clinical trial, no serious adverse events were observed when filociclovir was administrated to healthy volunteers [20]. A planed phase II clinical trial will provide more insights into the efficiency in solid organ transplant recipients.

Brincidofovir (Figure 2) is a lipid-conjugated prodrug of cidofovir [21]. This formulation led to increased oral bioavailability, higher intracellular concentrations of the active drug, and lower plasma concentrations of cidofovir [21]. Analysis revealed that brincidofovir has a higher antiviral activity in vitro compared to cidofovir [21]. In a phase 2 trial, it was shown that the compound prevented HCMV reactivation [22]. Despite having a great potential, brincidofovir recently failed in a phase 3 study evaluating its ability to prevent HCMV infection in hematopoietic cell transplant recipients [23]. All the studies with the oral application of brincidofovir have been terminated, due to adverse side effects (ClinicalTrials.gov accessed on 12 March 2021, NCT02439979; NCT02439957).

Maribavir is a benzimidazole L-riboside (1-(ß-L-ribofuranosyl)-2-isopropylamino-5,6-dichlorobenzimidazole; Figure 2) and an inhibitor of the HCMV pUL97 protein kinase [24,25]. Maribavir has several advantages over approved drugs: (i) oral bioavailability, (ii) up to 20-fold higher efficacy than GCV or CDV, and (iii) no cross-resistance to currently available drugs. In vitro, maribavir prevents viral replication in a dose-dependent manner. The 50% effective concentration ranges from 0.06 to 19.4 µM [26,27]. In addition, maribavir showed high antiviral activity during preclinical and clinical testing [28]. In a phase 2 trial, Chou et al. reported that, although a clearing of HCMV was successful (up to 77%), analysis of posttreatment samples of 23 patients showed the emergence of UL97 mutations that confer maribavir resistance [29]. The aim of a randomized phase 3 trial is the comparison of the efficacy of high-dose maribavir treatment (200 mg twice daily) with that of currently available drugs. The study group consists of patients who underwent hematopoietic stem cell or solid organ transplantation and are infected with HCMV that was refractory or resistant to first-line treatment (ClinicalTrials.gov accessed on 12 March 2021, NCT02931539). The upcoming results from the phase 3 trials will show whether maribavir exhibits promising profiles against HCMV infection in transplant recipients.

## 3. Novel Inhibitors Targeting Host-Cell Factors

### 3.1. Dispirotripiperazines—Blocking Entry

Dispirotripiperazines are tricyclic molecules with two quaternary, positively charged nitrogen atoms (or “spiro atoms”). These compounds were initially synthesized as potential anticancer agents. The discovery of the antiviral activity of these molecules, particularly against herpesviruses, led to extensive chemical synthesis and studies. A brief history of the development of these compounds is presented in our recent article [30]. Schmidtke et al. previously demonstrated that the dispirotripiperazine DSTP-27 (Figure 3, compound 1) not only effectively inhibits the replication of various herpes simplex virus strains, but also shows moderate activity against the HCMV AD169 strain, with an IC_50_ value of 12.3 µM [31]. The authors speculated that such a positively charged molecule might prevent the virus life cycle via binding to negatively charged cell-surface heparan sulfates. In a further study, the in vitro activity of DSTP-27 was found to be more potent towards laboratory strains and ganciclovir-resistant/sensitive clinical isolates (EC_50s_~0.15–0.85 µM), with no observed cytotoxicity in human embryonic lung fibroblasts [32]. The mechanism of the anti-HCMV action of DSTP-27 is not completely understood [32]. The results from time-of-addition assays demonstrated its effect on the early stage of HCMV infection. By contrast, attachment and penetration analyses revealed that virus attachment is not blocked. Moreover, DSTP-27 is known to be metabolically unstable, preventing its further development [33]. More recently, the antiviral properties of two novel dispirotripiperazines, 11826091 and 11826236 (Figure 3, compounds 2 and 3), were evaluated and described [34]. The compounds 11826091 (2) and 11826236 (3) are active against low-passage as well as GCV-resistant HCMV, with an EC_50_ range of 1.38–8.95 µM, and are completely noncytotoxic (CC_50_~416 and 321 µM, respectively). In summary, both substances were well tolerated and exhibited a strong dose-dependent antiviral activity. Detailed studies revealed that both small molecules block virus attachment but not penetration. The virus inhibition through the binding of the cell-surface heparan sulfate by these molecules has been further refined and confirmed [31]. Surprisingly, one derivative, 11826236 (3), has also been found to have an antiviral effect when added to cells after infection, and the observed event may be the subject of further studies. In addition, the antiviral activity was further confirmed for pseudorabies virus (PrV), a member of the Alphaherpesvirinae [34]. In summary, these findings demonstrate the high potency of 11826091 and 11826236 as new herpesvirus entry inhibitors.

### 3.2. Artemisinins Bind Vimentin

In the last few years, the antimalarial drugs artemisinins have been repurposed for the treatment of HCMV. Artesunic acid (Figure 3, compound 4), a semisynthetic derivative of artemisinin with improved aqueous solubility, was found to inhibit the replication of both HCMV laboratory strains (AD169 and Towne) and clinical isolates resistant to available antivirals (EC_50_s) [35]. Moreover, the compound is effective in an experimental animal model [36]. Roy et al. demonstrated that the molecule might bind to the filament protein vimentin in order to prevent HCMV replication [37]. Nevertheless, the clinical significance of artesunate is still unclear. Single patients and clinical trials with a small number of patients demonstrated that the results after the treatment of either hematopoietic stem cell transplant recipients or solid organ recipients were contradictory [38,39,40]. While the treatment of patients with a mild disease succeeded, the treatment of patients with a fatal disease was ineffective [39]. Further trials are required to assess the antiviral efficacy of artesunate and artemisinins in the treatment of HCMV disease in a transplant setting.

Recently, another derivative of artemisinin, artemisone (Figure 3, compound 5), has been identified. The authors demonstrated that these derivatives have an antiviral activity against HCMV strains in different cell lines. The 50% effective concentrations range from 0.22 to 1.46 µM, comparable to or superior to that of the control ganciclovir (EC_50_~0.40–2.08 µM) [41]. In addition, the compound inhibits clinical isolates that are ganciclovir- and cidofovir-resistant. However, the cytotoxicity of artemisone was higher than that of ganciclovir in different cell lines, leading to high selectivity indices. Oiknine-Djian et al. have suggested that the mechanism of action of artemisone might target the early phase of the viral replication cycle, but clearer evidence is needed. In a follow-up study, the authors described a strong in vitro synergistic effect of artemisone in combination with cidofovir, brincidofovir, and maribavir, and a moderate synergism with letermovir and ganciclovir [42]. Nonetheless, as noted by Haynes et al., the molecule has a low aqueous solubility [43] that will result in poor oral bioavailability. Further attempts are required to overcome this issue.

## 4. Novel Inhibitors Targeting Viral Proteins

### 4.1. Nucleoside Inhibitors

Several heterocycles have been used as bases in nucleoside structures. In particular, Kozlovskaya et al. [44] investigated the antiviral efficacy of phenoxazine-based nucleoside derivatives (Figure 3, 6). Analysis revealed only a moderate antiviral activity against the HCMV strains Davis and AD169 (EC_50s_~4 µM), comparable to that of ganciclovir (EC_50s_ ~1.06 and 5.99 µM, respectively) but lower than that of cidofovir (EC_50s_~0.3 and 0.51 µM, respectively) [44]. In addition, this molecule is cytotoxic for human embryonic lung cells (CC_50_~12.2 µM), like other compounds of this series. Therefore, the authors will use these compounds as a starting point for further structure optimization to reduce cytotoxicity.

Paramonova et al. [45] synthesized a set of (ω-p-bromophenoxyalkyl) uracil derivatives with various acetamide moieties. The authors demonstrated that a molecule with a dodecane-1,12-diyl linker (Figure 3, 7) has a potent antiviral activity against HCMV. Cells infected with the strain AD169 or Davis were treated with two compounds. The 50% effective concentration for AD169-infected cells was 0.8 µM, while infections with the strain Davis lead to an EC_50_ of 1.52 µM. Neither compound is cytotoxic, although the structurally related compound with a decane-1,10-diyl linker is.

### 4.2. Quinazoline Targeting HCMV Kinase

Hutterer et al. [46] identified two quinazoline-based compounds, Vi7392 (9) and Vi7453 (10) (Figure 3), with potent antiviral activity against the HCMV AD169-GFP strain (EC_50_~1.77 and 3.11 µM, respectively). These were comparable to the value for the previously analyzed quinazoline Ax7396 (EC_50_~3.40 µM) (Figure 3, compound 8) as well as that for ganciclovir (EC_50_~2.50 µM); however, maribavir has the most potent activity (EC_50_~0.12 µM) [46]. In a following extended study, HCMV laboratory and clinical strains including maribavir- and ganciclovir-resistant variants were also susceptible to the compounds Vi7392 and Vi7453 (Figure 3, compounds 9 and 10), with EC_50_ ranges of 0.96–2.10 and 1.80–3.31 µM, respectively. Although Vi7392 is slightly more effective, it has higher cytotoxicity than Vi7453 (CC_50_~70.93 and > 90 µM for most cells), with CC_50_ values from 22.98 to > 90 µM (only for one cell type). Such differences in the cytotoxic concentrations might be due to the presence of an iodine atom in the structure of Vi7392 (Figure 3, compound 9). It should be noted here that this atom is rarely used in drug design. The authors revealed that these quinazoline-core molecules exert an antiviral response by targeting the viral protein kinase pUL97 [46]. Nevertheless, the prospects for further development are not fully understood, since the related compound Ax7396 (Figure 3, compound 8) showed no statistically relevant anti-cytomegalovirus activity in a mouse model. It is important to obtain more insight regarding the structure–activity relationship. Recently, the same research group investigated the biological potential of quinazoline Ax7396/artemisinin hybrid molecules [47]. Two hybrid molecules had the most efficient anti-HCMV activity (EC_50_: 0.15–0.2 µM).

### 4.3. Inhibitor of the Small Terminase Subunit pUL89

The HCMV terminase plays a key role in replication and consists of the large terminase subunit pUL56, the small terminase subunit pUL89, and the subunit pUL51 [10,11,12,13,14]. Gentry et al. [48] provide an overview of all the known HCMV terminase inhibitors.

Wang and colleagues discovered that hydroxypyridonecarboxylic acid (Figure 3, compound 11), originally designed as an HIV RNase H inhibitor [49], inhibits the terminase subunit pUL89 nuclease activity (IC_50_ of 1–6 µM depending on the conditions) and reduces HCMV replication, with an EC_50_ of 4 µM. Moreover, compound 11 did not affect cell viability (CC_50_ > 200 µM) [49]. A further structure–activity relationship study revealed that the replacement of the fluorine atom with a bulky group, such as methylsulfonamidomethyl-phenyl, has a beneficial effect on the anti-UL89 and anti-HCMV activities [50]. Subsequent analysis demonstrated that the compound 12 (Figure 3) inhibits HCMV replication in cell culture, with an IC_50_ of 1.9 µM, which is higher than a previously obtained result (IC_50_~4.9 µM). In addition, this compound has a slightly higher antiviral activity against pUL89-C (IC_50_ of 5.6 µM) compared to 11 (IC_50_~6.2 µM) [50]. The authors hypothesized that a special core may be required for potent inhibition, and proposed a possible pharmacophore model structure with a chelating triad including a hydroxy group, a carbonyl group, and a carboxylic acid, as well as a mono- or disubstituted hydrophobic phenyl or biphenylmethyl substituent at N1. Furthermore, another structurally similar compound, 3-hydroxy-6-((4′-methyl-[1,1′-biphenyl]-4-yl)amino)pyridine-2,4(1H,3H)-dione (Figure 3, compound 13), was identified. This compound has a low IC_50_ value according to a pUL89-C nuclease immunoassay and a low EC_50_ value for the inhibition of HCMV replication (2.2 and 1.2 µM, respectively) [51]. The compound acts in the same way as those mentioned above.

An overview of the characteristics of the described novel small molecules is presented in Table 1.

## 5. Conclusions

Small molecules are the most promising compounds for the antiviral treatment of human cytomegalovirus. These compounds have significant advantages over currently available therapies. One advantage is the targeting of an early stage in infection (viral entry, nucleic acid synthesis, DNA packaging, or nuclear egress). Their high efficacy, leading to safe and potent antiviral agents, is a further advantage. Additional important considerations are the potential for large-scale production and stability. The latter is a prerequisite for broad distribution, a requirement for worldwide treatment. We provide insights into the structure–function relationships of the presented compounds and describe efforts that have been made to further optimize these small molecules. These encouraging data should inspire the continued development of small molecules as anti-HCMV drugs.

## Figures and Tables

**Figure 1 viruses-13-00474-f001:**
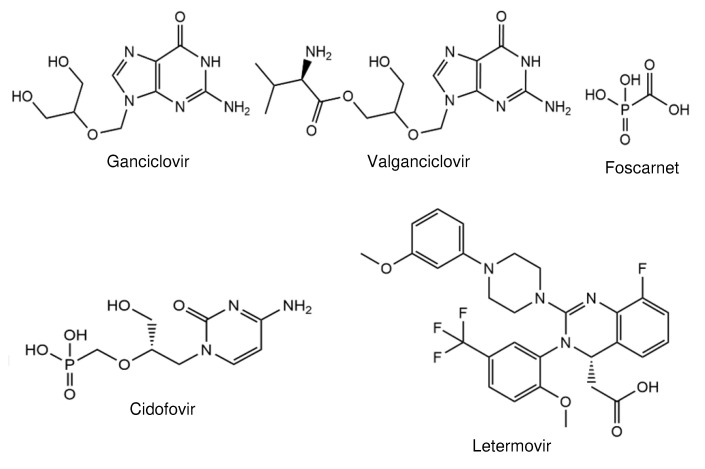
Structures of approved drugs against human cytomegalovirus (HCMV).

**Figure 2 viruses-13-00474-f002:**
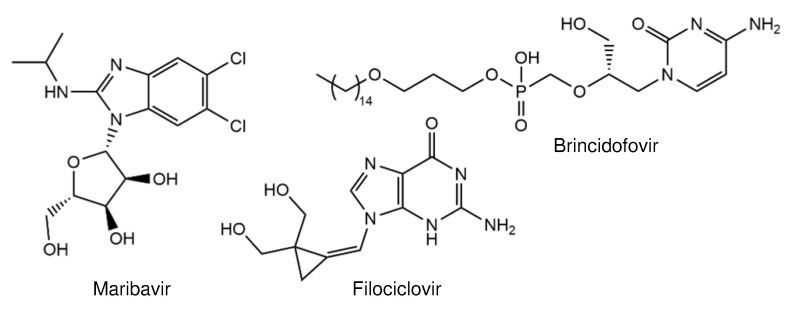
Structures of antivirals in clinical trials.

**Figure 3 viruses-13-00474-f003:**
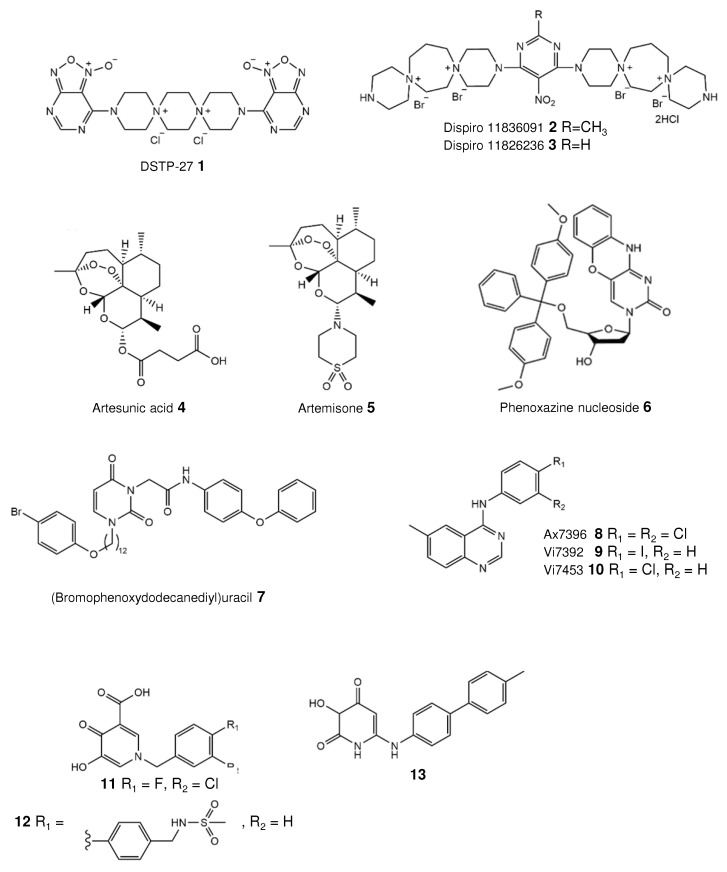
Structures of novel small molecule inhibitors.

**Table 1 viruses-13-00474-t001:** Overview of novel HCMV inhibitors.

Compound Name	IC_50_ (µM)	Mechanism of Action	Ref.
Dispirotripiperazines 11826091 and 11826236	1.38–8.95	Blocking virus attachment by binding cell-surface heparan sulfates	[31]
Artesunic acid	5–15	Binding of vimentin, a filament protein	[35,37]
Artemisone	0.22–1.46	Targeting an early phase of the viral replication cycle	[41]
Phenoxazine nucleoside 6	4	Nucleoside mimetic	[44]
(Bromophenoxydodecanediyl) uracil 7	0.8–1.52	Nucleoside mimetic	[45]
Quinazolines Vi7392 and Vi7453	0.96–3.31	Targeting viral protein kinase pUL97	[46]
Hydroxypyridonecarboxylic acid 11	4	Inhibition of the small terminase subunit pUL89	[49]
3-Hydroxy-pyridinedione 13	1.2		[51]

## Data Availability

The data presented in this study are available on request from the corresponding authors. The data are not publicity available due to restricted access to the servers of the Charité-Universitätsmedizin Berlin and of the Russian Academy of Science.

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
