# Peer review of "Small Molecules—Prospective Novel HCMV Inhibitors"

_viruses, 2021, doi:10.3390/v13030474_

Round 1

Reviewer 1 Report

This manuscript is a review entitled "Small Molecules-perspective novel HCMV inhibitors."  This is a nice concise overview/review of the topic of inhibitors for HCMV.  The only issue with the review is that throughout the manuscript "moderate" English Language changes are required.  As one small example, in the abstract on line 16: Replace: It is urgently needed to find…. With: There is an urgent need to find….  This is only one example of the many "language" issues that need to be edited before acceptance of the manuscript.

Author Response

Dear Reviewer,

Thank you for your kind review.

We edit the revised manuscript together with my colleague Terry Jones, a native speaker.  All changes are highlighted in the revised version. This increase the readability.

Kind regards

Elke Bogner

Reviewer 2 Report

This is a wonderful review of the new HCMV inhibitors but I missed seeing a table for them. A table would be very useful for the researchers and clinicians for the quick look and will increase the value of the paper.

Tables like these about inhibitors appear easily in the image search and over the course of time, help to increase the citation of the paper. e.g. Compound name, IC50, Mechanism of action, Reference study/studies, Comments can be included in the table!

Author Response

Dear Reviewer,

Thank you very much for your review. 

We follow your recommendation to include a table . This is a great idea to present the results in an overview. the table is included on page 7.

Further we perfomed spell check with my colleague Terry Jones. The changes are high-lighted in the revised manuscript.

 Elke Bogner

Round 2

Reviewer 2 Report

The manuscript looks much better and can be considered for publication!